# Automatic Process Comparison for Subpopulations: Application in Cancer Care ^[note 1]^

**DOI:** 10.3390/ijerph17165707

**Published:** 2020-08-07

**Authors:** Francesca Marazza, Faiza Allah Bukhsh, Jeroen Geerdink, Onno Vijlbrief, Shreyasi Pathak, Maurice van Keulen, Christin Seifert

**Affiliations:** 1Faculty of Electrical Engineering, Mathematics and Computer Science, University of Twente, 7522 NB Enschede, The Netherlands; f.a.bukhsh@utwente.nl (F.A.B.); s.pathak@utwente.nl (S.P.); m.vankeulen@utwente.nl (M.v.K.); 2Hospital Group Twente (ZGT), 7555 DL Hengelo, The Netherlands; j.geerdink@zgt.nl (J.G.); o.vijlbrief@zgt.nl (O.V.)

**Keywords:** process mining, process comparison, quality control, cancer types, breast cancer care, MIMIC database

## Abstract

Processes in organisations, such as hospitals, may deviate from the intended standard processes, due to unforeseeable events and the complexity of the organisation. For hospitals, the knowledge of actual patient streams for patient populations (e.g., severe or non-severe cases) is important for quality control and improvement. Process discovery from event data in electronic health records can shed light on the patient flows, but their comparison for different populations is cumbersome and time-consuming. In this paper, we present an approach for the automatic comparison of process models that were extracted from events in electronic health records. Concretely, we propose comparing processes for different patient populations by cross-log conformance checking, and standard graph similarity measures obtained from the directed graph underlying the process model. We perform a user study with 20 participants in order to obtain a ground truth for similarity of process models. We evaluate our approach on two data sets, the publicly available MIMIC database with the focus on different cancer patients in intensive care, and a database on breast cancer patients from a Dutch hospital. In our experiments, we found average fitness to be a good indicator for visual similarity in the ZGT use case, while the average precision and graph edit distance are strongly correlated with visual impression for cancer process models on MIMIC. These results are a call for further research and evaluation for determining which similarity or combination of similarities is needed in which type of process model comparison.

## 1. Introduction

The quality of health care can be assessed by observing the structure, processes, and outcomes of healthcare [1]. Processes in organisations, such as hospitals, may deviate from intended standard processes, due to unforeseeable events and the complexity of the organisation. For hospitals, the knowledge of actual patient streams for different patient populations (e.g., severe or non-severe cases) is important for quality control and improvement. Electronic health records (EHR) contain a wealth of information about patients, including timestamps of diagnoses and treatments. Thus, EHR can serve as input data to discover the as-is processes [2] in healthcare. In order to investigate patient processes for patient populations of interest (e.g., severe or non-severe cases), their process models have to be constructed and compared. Manual comparison of process models requires expertise in understanding process models and is time-consuming, which makes it unfeasible for many populations of interest. In this paper, we present an approach to automatically compare process models that were obtained from EHR event logs in order to have an initial quantification of the degree of similarity between different patient subgroups. More specifically, we compare three different approaches for obtaining a similarity between process models: (i) visual inspection, i.e., human judgment, (ii) cross-log conformance checking, and (iii) similarity measures on directed graphs extracted from the process model. For cross-log conformance checking, we apply the replay technique using the event log of one population against the process model that was discovered for a second population (and vice versa). Then, we evaluate which of the methods for measuring process similarities best approximate human judgment. The two cases that we considered are the publicly available MIMIC data (https://mimic.physionet.org/, last accessed on 25 February 2020) with the focus on different cancer patients in intensive care, and data on breast cancer patients from Ziekenhuis Group Twente (ZGT). In summary, the contributions of this paper are the following:We present process models for different cancer types obtained on MIMIC, and process models for breast cancer patients for different patient sub-populations.We show how process models obtained by inductive visual miner and Petri nets can be converted to general graphs.We propose methods to quantitatively compare process models, based on conformance checking and graph similarity measure.We performed a user study to obtain a ground-truth for visual similarity of process model pairs and evaluate our methods in the domain of cancer care in order to compare different patient populations.

This work extends own previous work [3], as follows: Additional to process model analysis for breast cancer sub-populations, we added an analysis for general cancer types on the publicly available MIMIC data base. Further, we show how to create general graphs from Petri-nets to obtain graph-based similarity metrics. Lastly, we strengthened the measured human-based visual comparison by obtaining visual similarity measures from a user study with 20 users.

In the remainder of this paper, we first introduce the application domain cancer care (cf. Section 2), review related work (cf. Section 3). We then present the approach in detail (cf. Section 4), and its evaluation for MIMIC cancer populations (cf. Section 5) and for breast cancer populations from ZGT hospital (cf. Section 6).

## 2. Application Background

Electronic health records (EHRs) can be a solution for improving the quality of medical care. EHRs represent digitally collected longitudinal data, such as medical reports, images, sensitive data and clinical information about patients and their provided treatments [4].

### 2.1. Cancer Types in MIMIC

MIMIC-III is a large, freely-available relational database comprising de-identified health-related data that are associated with over forty thousand patients who stayed in critical care units of the Beth Israel Deaconess Medical Center between 2001 and 2012 [5]. This database is the result of the medical records digitization and the open nature of the data allows clinical studies to be reproduced and improved in ways that would not otherwise be possible. We obtained access to the data by completing the required online course on data use and protection. The database encompasses a diverse and very large population of ICU (Intensive Care Unit) patients, including cancer patients. Cancer types are organised via the “The International Classification of Diseases, 9th Revision, Clinical Modification” (ICD-9 codes) (https://www.cdc.gov/nchs/icd/icd9cm.htm, last accessed on 25 February 2020) taxonomy, which knows 13 different types of cancer in the highest level in the hierarchy. We evaluate how different cancer types are treated in the ICU setting represented in the MIMIC database. For the analysis, we focus on the top level events in MIMIC: evaluation and management, anesthesia, surgery, radiology, emergency technology, performance measurement, and medicine. We chose to focus on the general events, because the second level of granularity contains 134 sub-events, leading to process models intractable for pairwise visual comparison.

### 2.2. Breast Cancer Populations for ZGT

From the patient database of the Hospital Group Twente (ZGT), we were interested in processes for different patient populations in the domain of breast cancer. The considered EHR for those patients contains four general types of reports, radiology, pathology, surgery, and multi-disciplinary reports. Radiology reports communicate the findings of imaging procedure by describing the radiology images (e.g., X-rays). In case of a patient with a suspicion of breast cancer, it also contains a BI-RAD score. Pathology reports are free medical texts where a diagnosis based on the pathologists examination of a sample of the suspicious tissue is given. The narrative operative surgery reports document breast cancer surgery. The multidisciplinary reports (MDO) are free-texts written during a multidisciplinary expert team meeting. One of the most important data included in the radiology report is the BI-RADS category. Breast Imaging-Reporting and Data System (BI-RADS) [6] is a classification system proposed by the American College of Radiology (ACR) to represent the malignancy risk of breast cancer of a patient in a standardized manner. A BI-RADS category can range from 0 to 6, with 0 being benign and 6 being the most malignant. Patients with different BI-RADS follow different processes e.g., patients with BI-RADS category 0 need additional imaging evaluation, BI-RADS category 3 needs initial short-interval follow-up, and BI-RADS category 4 may be recommended for biopsy. In Netherlands, women between the ages of 50 and 75 are solicited for screening once every two years. The purpose of the screening program is to detect the breast cancer at an early stage, before symptoms appear.

## 3. Related Work

**Quality in health care** and the corresponding reporting and evaluation is an issue of national and international importance. Donabedian [1] proposed that the quality of health care can be assessed by observing the structure, processes, and the outcomes of healthcare. The Institute of Medicine (IOM) defines health care quality as “the degree to which health services for individuals and populations increase the likelihood of desired health outcomes and are consistent with current professional knowledge” [7]. Process-based quality measures are more suited to explain how and what is required to improve health care processes, as compared to outcome-based measures [8,9]. Measuring the quality of various processes can also answer questions, like accuracy of diagnosis, disease monitoring and therapy, and percentage of patients, who received care as recommended [8]. Better process quality can also lead to better patient satisfaction with the series of transactions occurring during their hospital visit [10]. To summarize, the previous works state that quality of health care can be improved by measuring the health care processes, which can further lead to better health outcome.

The goal of **process mining** is to extract process models from event logs [11], also known as transactional logs or audit trails [12]. An event corresponds to an activity (i.e., a well-defined step in the process) affiliated with a particular case (i.e., process instance) [13] and particularly consist of a time stamp and optional information, such as resources or costs. Process mining as a discipline consists of three dimensions, process discovery, conformance checking and process enhancement [14]. *Process discovery* refers to the construction of a comprehensive process model, e.g., Petri-Nets or State-charts, to reproduce the behaviour seen in the log file [13]. We use the Inductive Visual Miner(IvM) [15] plug-in of ProM (promtools.org, last accessed on 1 May 2020), since it provides an user-friendly visualization, with an opportunity to investigate deviations. The deviations represent cases which do not follow the most common behaviours and thus correspond to event log traces that the process does not explain. *Conformance checking* is applied to compare process models and event logs in order to find commonalities and discrepancies between the modeled behavior and the observed behavior [16]. Our goal is to compare two processes; therefore, we use the (ProM) plug-in “Replay a Log on Petri Net for Conformance Analysis” to play the event log of one patient population to the discovered process model of another population and use the obtained fitness, precision and generalization measures for our similarity analysis. A case study explored the applicability of **process mining in health care** and raised the concern that traditional process mining techniques have problems with unstructured processes, as they can be found in hospitals [17]. In this paper, we will focus on a very small sub-domain, breast cancer care, in order to reduce the complexity of the EHR extraction and resulting process models.

Kurniati et al. [18] worked on process mining on oncology using a public dataset, MIMIC-III. Their work involved—(i) finding the most followed path and some exceptional paths taken by patients, (ii) finding the differences in the paths followed by different cancer types, and (iii) finding the activities with long waiting times. They used the top level granularity of events to create process models, consisting of events—admission, emergency department, ICU, discharge etc. They compared three process models generated from cancer types with the three highest cases. They evaluated their process models using fitness and precision scores and found that the most frequent events are admission and discharge. They also found, through visual inspection, that admission, discharge, ICU in, and ICU out are the common activities in each cancer type. In our paper, we use the same cancer types from MIMIC dataset to create process models, but for different activities and apart from visual inspection, we use cross-log conformance checking and similarity measure to compare the processes.

Even for a small set of EHR to have a useful insight into the patients’ health path, we need to compare extracted business processes models. Process models are traditionally compared with linear search techniques where a query model is compared to each model in the collection and focuses on process models that are composed of atomic tasks and connectors [19]. In a complex domain like health care, a lot of research attention has been paid to other process modeling constructs, such as sub-process invocation, exception handlers, control-flow view of process models, and resource allocation views [20]. In the health care domain, we need to differentiate between slightly different models and completely different models. Process model equivalence notions are not sufficient, because they produce a binary answer. In practice, the process models are compared with respect to some typical properties [21]. Becker et al., proposed 23 desirable properties for business process comparison. Experiments show that hardly any experiment fulfills all desirable properties [22]. Thus, there is not a single “perfect” business process model comparison method. In this paper, we investigate the applicability of similarity metrics proposed for general graphs (e.g., [23]).

When **comparing two graphs**
$G_{1}$ and $G_{2}$ one is either interested in exact matches of full or sub-graphs (graph homomorphisms) [24] or a measure of structural similarity (e.g., [23,25]), among which the graph edit distance (GED) [24] is most widely adopted. The GED is defined as the minimum number of operations (add/remove/substitute nodes and edges) needed to transform $G_{1}$ to $G_{2}$. The problem of calculating the GED is NP-hard in general, which makes it unfeasible to solve for larger graphs and giving rise to heuristic approximation approaches [26]. More recently, supervised machine learning approaches have been suggested. For instance, Li et al. use a combination of two neural networks to learn a similarity score for $G_{1}$ and $G_{2}$ [27]. As the authors demonstrate, supervised machine learning approaches can generate highly accurate similarity scores, but they require ground-truth graph-similarity data for training the models. In this work, a graph-similarity ground-truth is not available; therefore, we use approximations of GED and an unsupervised machine learning approach based on hand-crafted features for $G_{1}$ and $G_{2}$ and standard distance metrics on these features.

## 4. Approach

In this work, we are interested in the care processes followed by different patient populations and how these processes compare to each other. In this section, we describe our approach for obtaining and comparing process models from EHR.

Figure 1 provides an overview of the approach.

The available data are EHR, extracted and anonymized from a patient database. Subsequently, event logs are constructed for populations of interest, the corresponding process models are constructed (cf. Section 4.1) and transformed to directed graphs (cf. Section 4.2). Two process models are then compared by (i) visual inspection using obtaining a similarity measure based on human judgment, (ii) cross-log conformance checking, and (iii) graph comparison methods (cf. Section 4.3). The approach is performed on the general cancer types from MIMIC-II and on ZGT breast cancer populations.

### 4.1. Process Discovery from EHR

In a first step, the EHR of patient subgroups of interest are extracted from the hospital data base. After defining the events of interest, only information related to those events are extracted from the EHR. Depending on the hospital database, this process involves a combination of hand-crafted filtering and extraction rules. Additionally, personal data have to be anonymized. In order to apply process mining techniques, EHRs are transformed to event logs. Each event is associated with a case, i.e., a patient, and events are chronologically ordered. Each case can have multiple events. Based on the question of interest, patient data can be divided in different ways, creating various populations of interest. For each subpopulation, the event log and the process models is generated using the Inductive Visual Miner [28], which is also capable of detecting deviations. In our setting, deviations are patients whose behaviour differs from the most frequently observed paths.

### 4.2. Graph Construction

We converted the process models to directed graphs in order to use graph comparison methods.

*Graph Construction from Petri Nets:* petri nets consist of transitions, places and events. The edges are directed and unweighted. We construct directed (unweighted) graphs G(V1,V2,V3,E) from Petri nets as follows: Nodes in V1, V2 and V3 represent places, transitions and events, respectively. Edges between the respective nodes in the Petri net are added to the set of edges *E*.

*Graph Construction from IvM Models:* process models discovered with the IvM plugin consist of nodes, with an associated weight and directed, weighted edges. These process models were translated to directed, weighted graphs G(V1,V2,E), as follows: nodes in V1 represent activities and nodes in V2 correspond to logical operators in the process model. A directed edge is inserted between two nodes if it in between activities or activities and operators in the underlying process model. We used Boolean function operators to capture the semantic meaning of the process model in the directed graph. Operators can either be AND or XOR, the addition of “-split” or “-join” indicates the start and end of the respective paths. Thus, an “AND-split” means that patients have to follow both paths that the operator outlines, without particular order. The “AND-join” indicates the end of the paths that must be executed in parallel. The XOR operators work similarly, but states that the patient takes either one path, but not both. “Loop” indicates a cycle in the process. Each edge has an associated weight that is set to the frequencies of the connection in the process model.

### 4.3. Process Model Comparison

The reason to strive for automatic comparison of process models is the following: we define a potential large number of interesting patient sub-populations. Depending on the number of events of interest, the process models become very large and understanding the processes and their differences is a complex task in itself. We aim for grouping of process models along the questions “What are very similar patient groups” and “Which patient groups take very different paths through the hospital?” in order to support efficient and effective initial assessment. To facilitate such grouping, a similarity measure between process models is required that reflects the human-judged similarity. Therefore, we use human judgment as ground truth proxy and evaluate different similarity measures against this ground-truth. Figure 2 provides an overview of the approaches for process model comparison.

#### 4.3.1. Visual Similarity Assessment

For visual similarity assessment, pairs of process models were judged by humans for their similarity. We used a five-point Likert scale (0: Identical, 1: Slightly Different, 2: Somewhat Different, 3: Very Different, 4: Extremely Different). We performed a user study with 20 users (11 male, 9 female; aged 20 to 50 years). All of the participants have at least a University bachelor degree, in particular 11 of them have a master degree, five a bachelor degree, and four a PhD. Participants judged pairs of process models independently, and they were given the instruction to focus on the structure of the process and not on the numbers of the edges when assessing the similarity. The user groups were the same for both experiments (MIMIC and ZGT data).

#### 4.3.2. Cross-Log Conformance Checking

We used the (ProM) plug-in “Replay a Log on Petri Net for Conformance Analysis” to play the event log of one subpopulation with the process model generated by the second subgroup (and vice versa) and recorded standard conformance checking metrics: fitness, precision, and generalization.

#### 4.3.3. Graph-Based Comparison

For graph-based comparison of process models, we used the networkx graph library for python for calculation graph metrics and similarities (https://networkx.github.io/, last accessed on 25 February 2020). The graphs constructed from IvM models are weighted, whereas the graphs that are obtained from Petri nets are unweighted graphs (see Section 4.2). For both graph types, we calculate the graph edit distance (GED) using an approximation algorithm [29]. The GED function requires a function deciding when two nodes are considered a match, i.e., when two nodes are considered equal for the purpose of graph comparison. We would like to compare the structure of two graphs, but also retain some semantics for graph comparison. For this reason, we only consider a pair of nodes as a match, if they belong to the same type. In the following, we detail the types of nodes for both, IvM based graphs and Petri-Nets.

Node types of IvM based graphs are events and branching nodes that represent logical functions, like AND or XOR, or indicating loops. Event nodes match, if they are identical, whereas other nodes match if they have the same type. Consider the example in Figure 3, left. Graphs $G_{A}^{1}$ and $G_{A}^{2}$ have the same structure and node types and their event “Radiology” matches. Their only difference are the ids of the non-event nodes. Thus, the GED of $G_{A}^{1}$ and $G_{A}^{2}$ is 0. $G_{A}^{3}$ has the same structure and types of non-event nodes, but a different event “Surgery”. The GED of $G_{A}^{2}$ and $G_{A}^{3}$ is 1, since the 1 operation, i.e., a node substitution of “Surgery” with “Radiology”), is needed in order to match the graphs. The principle of node matching is similar for Petri nets, as shown in Figure 3, right. Node types of Petri-net based graphs are transitions, places, and events. Transition and place nodes match if they have the same type. Event nodes only match if they represent the same event. $G_{B}^{1}$ and $G_{B}^{2}$ only differ in the node ids of transitions and places, thus their GED is 0. Similarly to $G_{A}^{2}$ and $G_{A}^{3}$, $G_{B}^{2}$, and $G_{B}^{3}$ require one operation to be transformed into the respective other graph, resulting in a GED of 1. We ran the GED approximation for one hour for all graph pairs and report the minimum value obtained within this time frame.

For feature-based comparison of two graphs, we generated a feature vector for each graph. For the weighted directed graphs that were obtained from IvM models, we used the following measures: number of nodes, number of edges, average degree, average weighted degree, average clustering coefficient, average shortest path, average closeness, and average betweenness centrality. For the unweighted directed graphs obtained from Petri nets, we omitted the average weighted degree and the cluster coefficient. The former is equal to the average degree and the latter is 0 for all graphs constructed from Petri nets. We then obtained the similarity score for two graphs by first normalizing the feature vectors to unit length and then calculating the Euclidean distance of the two normalized vectors. The similarity score is then one minus the obtained distance.

## 5. Experiments on Cancer Types in the Public MIMIC Data Base

In our experiments, we focused on eight general events in order to compare processes for 13 different cancer type populations. First, we extract the event log by applying the selection criteria to collect records of treatment cases for patients diagnosed with cancer (cf. Section 5.1) and inspect the resulting patient populations (cf. Section 5.2). We then generate the process models using the Inductive Miner Plugin (cf. Section 5.3).

### 5.1. Event Log Preparation and Event Structure

The MIMIC database consists of 26 tables, linked by identifiers that usually have the suffix ‘ID’. We use the CPTEVENTS table, which contains current procedural terminology for patients, representing a distinct procedure performed on the patient during their ICU (Intensive Care Unit) stay. In the CPTEVENTS table, the information about the event’s date and the sequential ordering of events in a day (column “ticket_id_seq”) are used to obtain the final event sequence. In total, we found eight different types of events: evaluation and management, surgery, radiology, anesthesia, emerging technology, pathology and laboratory, performance measurement, and medicine. These types were assigned using the D_CPT table, which also lists sub-events for each top-level event. We only use the first level of granularity given the large number of sub-events (134 in total).

### 5.2. Patient Populations

In the second step, we selected records of patients diagnosed with cancer. We used the provided International Classification of Diseases, Clinical Modification (ICD-9-CM) codes to identify cancer patients. Cancer patients have a four-digit ICD code ranging from 140 to 239 in the MIMIC-II table DIAGNOSED_ICD table. An overview of the cancer types and the number of patients and associated events can be found in Table 1. In total, 7361 distinct patients were found to have at least one diagnose related to cancer. A patient might have more than one type of cancer, which make this patient fall into more than one group. The majority of cases belong to cancer type 7 (2620 patients), type 2 (1295 patients), and type 10 (1066 patients) and type 8 (1046 patients). For cancer type 8, we observe the highest number of events per case (39.3 events per case on average). To investigate the patient overlap between these populations, we calculated the Jaccard coefficient, as follows: for each population, we created a set with anonymized patient ids. The Jaccard coefficient is the ratio of the number of patients two populations have in common (set intersection) and the total number of unique patient ids (set union). As shown in Table 2, most patient populations have nearly no overlap (Jaccard similarity ≤0.03), while Type 2 and Type 7 show some overlapping patients (Jaccard similarity 0.16).

### 5.3. Process Discovery and Graph Construction

The process models for the cancer types with most cases are depicted in Figure 4 (type 2, 7 and 8), the remainder is shown in Section A.1. For this experiment, we used a Petri Net representation through inductive miner algorithm. Because of unclear software issues, we were not able to create a process model with IvM plug-in of ProM (as we did for ZGT data). For this reason, we choose the Petri net representation for MIMIC. Figure 5 shows the process model and the constructed graph for cancer type 10.

### 5.4. Process Model Comparison

Table 3 summarizes the comparison of the process models. For visual assessment, we report the average similarity score and standard deviation obtained by human judgment on a five-point Likert scale (0: identical to 4: extremely different). Additionally, we report fitness and precision obtained by cross-log conformance checking and their averages. We also report the GED and the similarity obtained by feature-based graph comparison (FS).

To analyse the relation between the similarity measure, we visualize univariate distributions, the pair-wise scatter plots, and calculate the corresponding Spearman’s rank correlations $\rho_{s}$ and associated *p*-values (mentioned in brackets under the $\rho_{s}$). We use $\rho_{s}$ to assess the monotonicity of the relationship between any two similarity measure with +1 and −1 indicating perfectly monotonically increasing and perfectly monotonically decreasing relations, respectively. Figure 6 provides an overview of the results for the aggregated values of Table 3. Detailed results for all non-aggregated values can be found in Section A.2. The diagonal plots show the normalized histogram for all of the similarity measures, and the estimated Gaussian kernel density distribution. The upper triangle shows $\rho_{s}$ (with corresponding *p*-values), and the lower triangle shows pairwise scatter plots with a linear regression line with the corresponding 95% confidence interval (shaded area).

Figure 6 shows high correlations between visual judgement and both, FS ($\rho_{s}=-0.94$, *p*-value = 0.01) and average fitness ($\rho_{s}=-0.89$, *p*-value = 0.02). More precisely, both FS and average fitness are negatively correlated to the visual judgement. The high correlation is supported by pair-wise scatter plots of average fitness vs. visual and FS vs. visual. It can be seen that the data points in the associated scatter plots lie close to the linear regression line with a reasonably small confidence interval band. Another significant observation is the high positive correlation of average fitness with FS ($\rho_{s}=0.81$, *p*-value = 0.05). However, it should be noted that all of the FS values are close to 1 and only range between 0.97 and 1.00. Further, we would have expected a positive correlation between FS and visual judgements—if features are similar, then the graphs should be visually similar. FS, therefore do not seem useful for comparing the underlying process models in practical settings. In conclusion, our results on the MIMIC dataset suggest that average fitness is an appropriate measure for quantifying visual judgement on graph similarity for the tested process model pairs.

## 6. Experiments on Breast Cancer Subgroups at ZGT

In our experiments, we collected EHRs from the hospital database (cf. Section 6.1), designed the event structure (cf. Section 6.2), created event logs for populations of interest (cf. Section 6.3), and obtained process models for these populations (cf. Section 6.4). We then compared these process models pairwise, by (i) visual inspection, (ii) cross-log conformance checking, and (iii) graph-based similarity measures and investigate how the obtained similarity measures reflect the similarity obtained by human judgment (cf. Section 6.5).

### 6.1. EHRs Extraction and Event Log Preparation

Free-text reports on breast cancer patients from 2012–2018 were collected from the hospital database. The following rules defined whether a patient was included in the analysis: with the purpose of identifying the complete health path of the patients inside the hospital, a patient has to be a “new patient”: in the range of time considered, the patient must have a first visit, which is not described in the referring report by keywords like “MRI”, “punctie” (biopsy), “mammatumor” (tumor in the breast). Also, it is not represented by a MDO or Surgery report, because it is impossible that the first visit of a patient is one of them. Accordingly, the start event is the first report for each patient that complies with these exclusions. We could not define clear criteria for determining an end event when a patient has finished the treatments in the hospital. Therefore, there is no end event in health path analysis. The gathered patients are 12,220, for a total of 44,219 reports.

### 6.2. Event Structure

We collected patient events on two levels of detail as shown in Figure 7. In the first level of granularity, events correspond to reports: radiology, pathology, MDO, and surgery reports. In the second level of granularity, data extraction and transformation techniques are applied in order to select and group event types. Starting from each sub-category, seven levels are finally established. The most complex (radiology event) includes 55 different procedures, some really different, others the same but written differently: they are collapsed in three sub-levels, diagnostic, biopsy, and other. “Diagnostic” refers to non-invasive imaging scans to diagnose a patient. “Biopsy” is a minimally invasive procedure that consists of taking a small piece of tissue to be analysed. “Other" groups the exceptions that are not included in the other two (few cases). Pathology is further divided into cytological and histological. Surgery report are divided based on the malignancy of the tumor. Finally, MDO is not further divided, hence is kept in the second level of granularity with the same label.

### 6.3. Patient Populations

We considered 6 different patient populations. We used an age threshold of 50 because of a known empirically established increased risk for breast cancer development at this age.
**SVOB:** patients coming from a national breast cancer screening program;**NoSVOB:** patients sent to the hospital by the general practitioner;**Birad12:** patients with a BIRAD score 1 or 2 (0% likelihood of cancer);**Birad3-6:** patients with a BIRAD score of 3 (probably benign), 4 (suspicious), 5 (highly suggestive of malignancy) or 6 (known biopsy-proven);**Age ≥ 50:** Patients of age 50 or older; and,**Age < 50:** Patients younger than 50.

Table 4 provides an overview for the selected populations showing the number of events with a breakdown on event type, i.e., the specific type of report, and the number of cases (patients). For each population, the event log contains at least 12,000 events, with radiology reports being the most frequent and surgery being the least frequent event type in all populations. Note that the radiology report is always present for each case. This result will be graphically confirmed by process figures where an AND condition is always existing for radiology events. Surgery events are generally occurring less frequently than the others event types. In particular, the percentage of surgeries for Birad12 and Birad 3-6 populations are significantly different. We also compare sub-populations on the second level of granularity only based on BI-RADS score. Table 5 shows the number of the events similarly to Table 4, but with a breakdown on event type on second level of granularity. The event distribution on this level is uneven. In all cases, one type is more frequent than the others.

We compared process models of populations that are of clinical interest, namely screening vs. non-screening patients (SVOB/NoSVOB), low vs. high probability of malignancy both in first and second level of granularity (Birad12 vs. Birad3-6, 2-Birad12 vs. 2-Birad3-6) and age groups before and after screening age (Age ≥50 vs. Age <50). We also included a comparison of patients with an age that do not require screening and non-screening patients (NoSVOB vs. Age <50) and screening patients that were transferred to the hospital, but had a low probability of malignancy (SVOB vs. Birad12). Table 6 provides an overview of the compared groups and shows the Jaccard similarity for different population pairs. As can be seen, there is a 30% overlap between NoSVOB patients and patients younger than 50, as well as between SVOB patients and patients with low likelihood of cancer (Birad12).

### 6.4. Process Discovery and Graph Construction

Process models obtained by the IvM plug-in of ProM (noise filtering set to 90%) with first level events for four example populations are shown in Figure 8 and Figure 9 (top). The models for the remaining population can be found in Section B.1. Blue rectangles represent events, the color intensity represents the amount. The process models for SVOB and Birad12 have the same structure (but different frequencies), although their patient populations are not the same (Jaccard similarity of 0.42, cf. Table 6). The process model for Birad3-6 patients has a complex structure, which indicates that patients with non-zero probability of malignancy follow quite complex paths in the hospital. Many deviations, represented by red dashed lines, exist in all processes.

Figure 9 shows the process model for NoSVOB patients and the correspondent translated graph. Circles denote the logic operators while rectangles represent events. The un-normalized feature vector *f* obtained from the graph for NoSVOB patients is f=(18,28,2.89,3.09,0.19,0.81,0.13,0.03), the features are in the order, as described in Section 4.3.

Figure 10 shows the process models for 2-Birad12 and 2-Birad3-6 patients with more fine grained events. We have twice as much event types on the second level, and the processes become clearly more complex. The model generated for patients with 2-Birad3-6 (3365 cases) is more complex than the other one, even though the patient population for 2-Birad12 (8612 cases) more than twice as large. Figure 10a shows no loop conditions and deviations are significant, similarly to the process that was generated for the same sub-population with only first-level events (cf. Figure 8b). This means that there are many patients that do not follow the most common behaviour. In Figure 10b, one rare event is missing (benign neoplasma mamma).

### 6.5. Process Model Comparison

Table 7 summarizes the comparison of process models. For visual assessment, we reported the average similarity score and standard deviation obtained by human judgment on a five-point Likert scale (0: identical to 4: extremely different). Further, we reported the fitness and precision values obtained by cross-log conformance checking and their averages.

We also reported the GED and the similarity obtained by feature-based graph comparison (FS). As the FS values are generally all above 0.9 and similar, they do not seem useful for comparing the underlying process models. Fitness and precision varies more and interestingly differs depending on whether the log of group 1 is played on the process model of group 2 or vice versa.

Figure 11 provides an overview of the results for the aggregated values of Table 7. Detailed results for all non-aggregated values can be found in Section B.2. Please refer to Section 5.4 for details of how to read the figure. Average precision shows the highest negative correlation with visual judgment ($\rho_{s}=-0.77$, *p*-value = 0.07), while GED shows the highest positive correlation with the visual judgement ($\rho_{s}=0.70$, *p*-value = 0.12), indicating human judgement to be aligned with GED. We also observe that a positive correlation between average precision and average fitness ($\rho_{s}=0.64$, *p*-value = 0.17) and FS ($\rho_{s}=0.67$, *p*-value = 0.15). Thus, the results show that GED and average precision are appropriate indications of visual similarity for this dataset.

## 7. Discussion

We addressed the problem of automatically comparing process models of different patient sub-populations in cancer care. Our experiments were performed in two related, but different applications areas (cancer types in ICU and breast cancer care) on two different data sets. We consider our work as a first step towards a solution for fully automatically comparing process models in order to alleviate manual effort. In this section, we discuss methodological choices and limitations of our work.

**Human Judgments.** Our user study comprised 20 participants each rating 6 (MIMIC) + 6 (ZGT) pairs of process models. The visual judgement scores showed a rather high standard deviation for most cases (cf. Table 3 and Table 7), except in the case for the two process that were exactly equal (SVOB and Birad12 on ZGT). In a preliminary study, in which four of the authors visually assessed the same process models independently, we observed smaller differences between assessors. This indicates that process model similarity scoring is a difficult task for untrained humans. Future experiments on artificially constructed process models in a more general domain, which does not require domain knowledge, could evaluate the validity of the human ground truth for process model comparison.

**Similarity measures.** We compared process models using (i) human visual assessment, (ii) graph-based measures and (iii) cross-log conformance checking. We assume that humans compare graphs by considering both, graph structure as well as global and local semantics. We leave a qualitative investigation for future work. For the graph based measures, GED mostly focuses on graph structure, but incorporates some semantics by only considering nodes of the same types as matches. Our feature similarities, on the other hand only consider the structure of graphs. Future work might explore semantic embeddings [30] as features for graph comparison. Cross-log conformance checking replays event data from one population on the process model constructed from events of a different population. Similarity based on cross-conformance metrics can thus be considered a semantic comparison, since event types and their sequences are considered.

**Approximation of GED.** While the GED provides the highest positive correlation with the visual judgement on ZGT data, the results need to be judged with care. We used an iterative algorithm for calculating the GED, and stopped the approximation after 1 h of calculation if the values did not change anymore. This means, that reported values might not be the true GED, as exemplified in the case of SVOB–Birad12 comparison on ZGT data, where GED should be 0 (cf. Figure 8 and Table 7). Additional research is needed to improve the computational efficiency for applications in practice.

**Validity of Results.** Different notions of equivalence (e.g., trace equivalence), produce binary answers (i.e., two processes and models are either equivalent or not). Such binary results are not very beneficial, especially in the complicated domains (such as health care), where we need to distinguish between marginally different models and entirely different models. To address the problem, we evaluated three different approaches (i) visual inspection, (ii) cross-log conformance checking, and (iii) graph-based comparison. We evaluated on two different application domains (general cancer types on MIMIC and breast cancer on ZGT) and used two different types of process models: Petri nets on MIMIC data and IvM based process models in the ZGT experiment. In each experiment, we compared six pairs of models, amounting to six data points per similarly measure per process model type. Thus, the underlying data for the analysis are quite limited. We observe different results on both datasets. For MIMIC, average fitness is an appropriate measure for quantifying visual judgement. For ZGT, GED and average precision are appropriate indications of visual similarity. Therefore, we would like to emphasize that our results are a first indication, and they can serve as a starting point for further investigations. However, further experiments are needed to verify the applicability and generalizability of the results to more process models pairs of the same data sets, different types of process models, other application domains, or more complex process models.

## 8. Summary

In this paper, we raised the problem of automatic quantitative comparison of process models generated from event logs with the same types of events. We proposed comparisons that are based on cross-log conformance checking and standard graph similarity measures obtained from the directed graph underlying the process model. We have contributed several metrics for subpopulation process model comparison that provide good indications of (dis)similarity. The experiments with two real-world data sets show that subpopulation process model comparison is a complex problem and further research is needed for determining which metric or combination of metrics is to be used in which situations. In our application scenario, the compared process models were rather small. The majority of process models on ZGT contained four event types, some had eight. The process models on MIMIC contained six different events. At a higher level of event granularity, process models easily get more complex and, thus, automatic comparison becomes more desirable.

## Figures and Tables

**Figure 1 ijerph-17-05707-f001:**
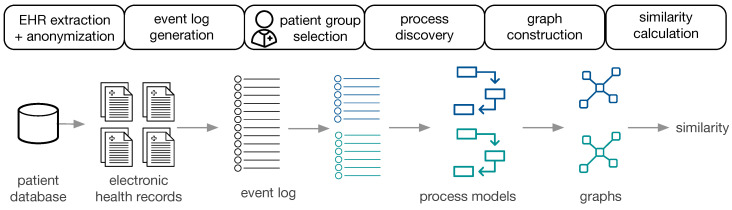
Overview of the approach.

**Figure 2 ijerph-17-05707-f002:**
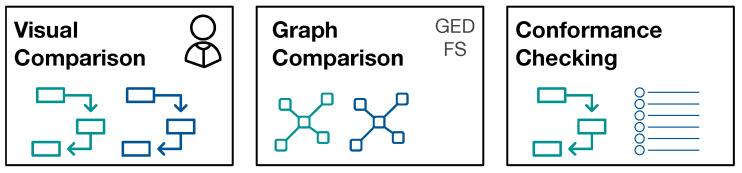
Overview of methods for pairwise comparison of process models.

**Figure 3 ijerph-17-05707-f003:**
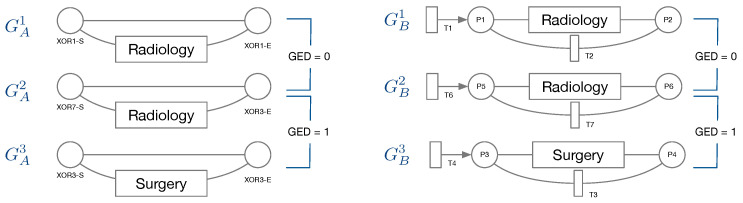
Intuition for graph-based similarity. Left: for graphs constructed from IvM models; right: for graphs constructed from Petri nets. Event nodes match only if they represent the same event. Other nodes match if they have the same type, e.g., represent the start of an XOR branch (XOR-S), independent on the ID of the graph node.

**Figure 4 ijerph-17-05707-f004:**
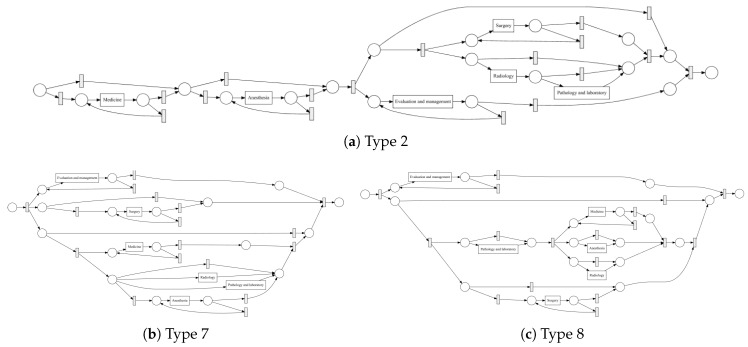
Discovered process models for cancer types 2, 7, and 8 in MIMIC.

**Figure 5 ijerph-17-05707-f005:**
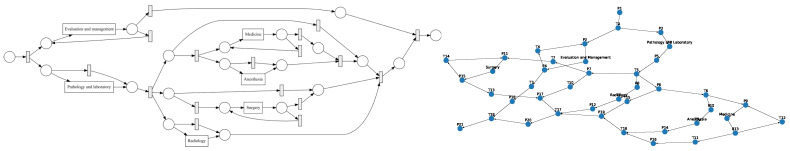
Process model (left) and constructed directed graph (right) for cancer type 10. Graph on the right was drawn using the networkx implementation of Kamada-Kawai path-length cost-function. Note that there is node and edge overlap for all event nodes, making the position nodes hardly visible.

**Figure 6 ijerph-17-05707-f006:**
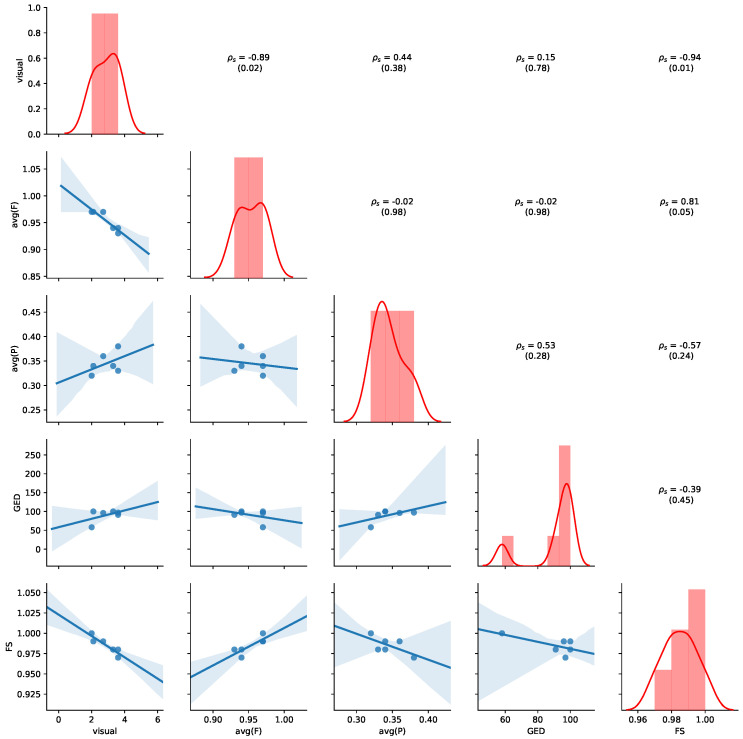
Relation of aggregated measures for MIMIC. Based on measures reported in Table 3. Diagonal: normalized histogram and kernel density estimation of the distribution. Lower triangle: scatterplot with estimated linear regression line. Upper triangle: pairwise Spearman’s rank correlations ($\rho_{s}$) with *p*-values.

**Figure 7 ijerph-17-05707-f007:**
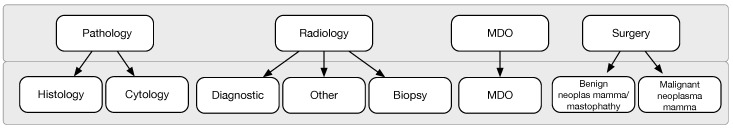
Event types on different levels of granularity (level 1 on top, level 2 at the bottom)

**Figure 8 ijerph-17-05707-f008:**
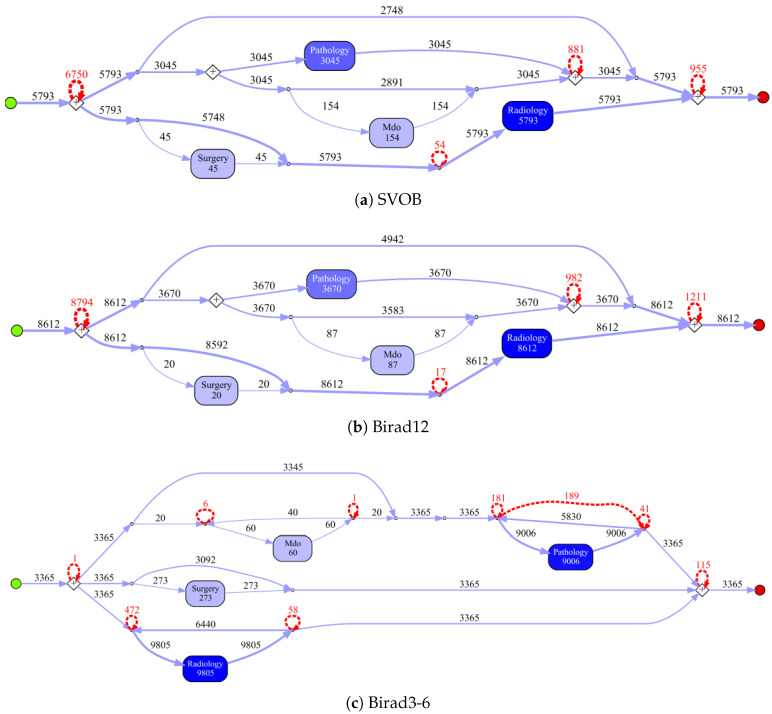
Discovered process models on ZGT for sub-populations with events on level 1.

**Figure 9 ijerph-17-05707-f009:**
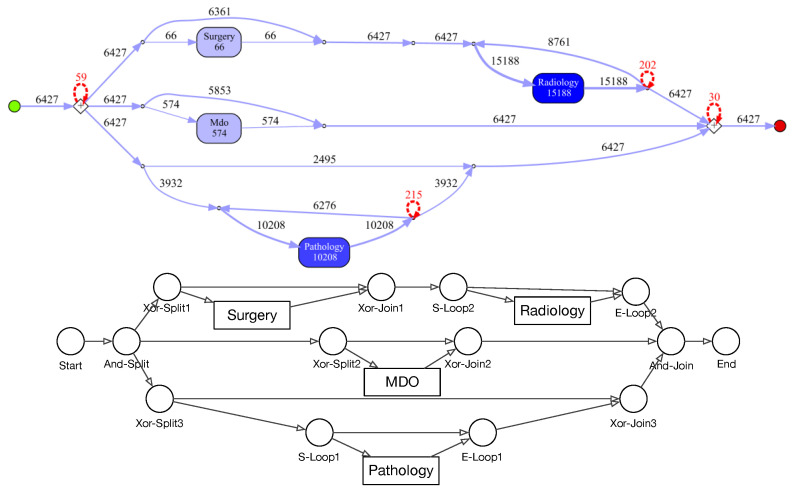
Process model for NoSVOB population (top) and derived directed graph (bottom). Edge weights and loops omitted in the graph for readability.

**Figure 10 ijerph-17-05707-f010:**
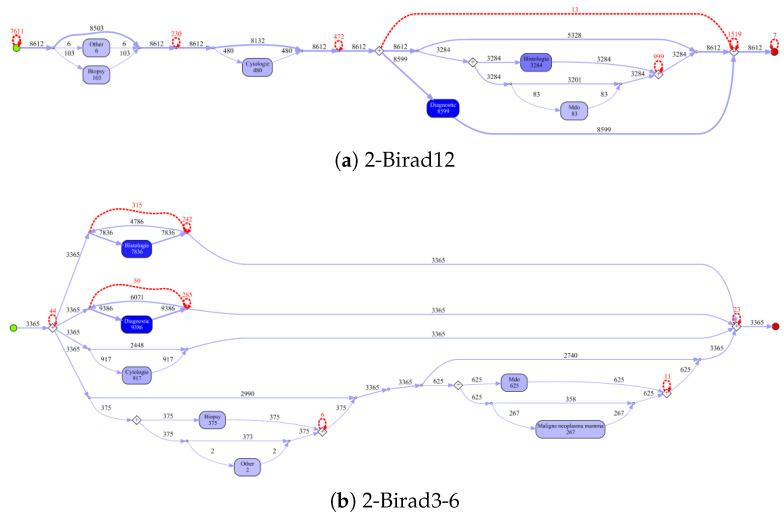
Discovered process models on ZGT for sub-populations with second-level events.

**Figure 11 ijerph-17-05707-f011:**
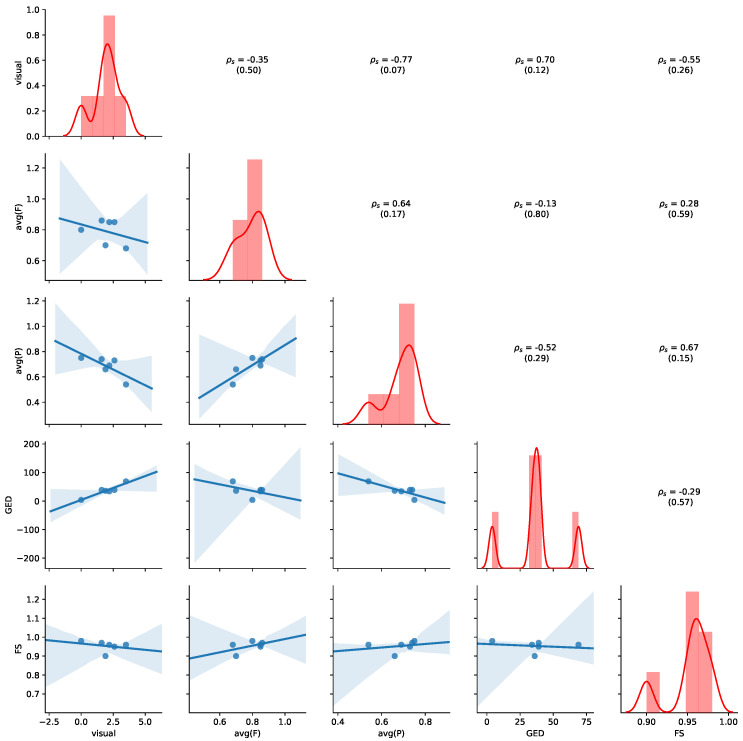
Relation of aggregated measures for ZGT. Based on measures reported in Table 7. Diagonal: normalized histogram and kernel density estimation of the distribution. Lower triangle: scatterplot with estimated linear regression line. Upper triangle: pairwise Spearman’s rank correlations ($\rho_{s}$) with *p*-values.

**Table 1 ijerph-17-05707-t001:** Patients with different cancer types. Patients can have multiple types of cancers. Bottom row in the table shows unique patients with any (one ore more) type(s) of cancer. The number in brackets is the associated ICD-9 code.

No.	Description	Cases	Events
1	Malignant neoplasm of lip, oral cavity, and pharynx (140–149)	82	1778
2	Malignant neoplasm of digestive organs and peritoneum (150–159)	1295	23,377
3	Malignant neoplasm of respiratory and intrathoracic organs (160–165)	982	18,225
4	Malignant neoplasm of bone, connective tissue, skin, and breast (170–175)	248	4713
5	Kaposi’s sarcoma (176–176)	13	296
6	Malignant neoplasm of genitourinary organs (179–189)	661	11,162
7	Malignant neoplasm of other and unspecified sites (190–199)	2620	43,275
8	Malignant neoplasm of lymphatic and hematopoietic tissue (200–209)	1046	41,139
9	Neuroendocrine tumors (209–209)	25	543
10	Benign neoplasm (210–229)	1066	23,974
11	Carcinoma in situ (230–234)	35	716
12	Neoplasms of uncertain behavior (235–238)	630	20,278
13	Neoplasms of uncertain nature (239)	57	1556
	Any type of cancer (140–239)	7361	

**Table 2 ijerph-17-05707-t002:** Population comparison. Reporting Jaccard similarity of sets of patient ids in the two groups.

Group 1	Group 2	Jaccard Similarity
Type2	Type7	0.16
Type2	Type8	0.01
Type2	Type10	0.03
Type7	Type8	0.01
Type7	Type10	0.02
Type8	Type10	0.02

**Table 3 ijerph-17-05707-t003:** Process comparison on MIMIC. Visual impression (Visual), conformance checking metrics (F—Fitness, P—Precision) when group 1 event log is compared with the process model (PM) constructed for group 2 and vice versa. Graph similarities (GED—graph edit distance, FS—feature-based similarity).

Populations			Conformance Checking						Graph Sim.	
			Log1-PM2		Log2-PM1		Average			
Group 1	Group 2	Visual	$F_{12}$	$P_{12}$	$F_{21}$	$P_{21}$	$\bar{F}$	$\bar{P}$	GED	FS
Type 2	Type 7	3.6±0.7	0.96	0.37	0.92	0.38	0.94	0.38	97	0.97
Type 2	Type 8	3.6±0.8	0.94	0.37	0.91	0.28	0.93	0.33	91	0.98
Type 2	Type 10	3.3±0.8	0.96	0.37	0.91	0.31	0.94	0.34	100	0.98
Type 7	Type 8	2.1±0.9	0.95	0.40	0.98	0.28	0.97	0.34	100	0.99
Type 7	Type 10	2.7±0.7	0.97	0.40	0.97	0.31	0.97	0.36	96	0.99
Type 8	Type 10	2.0±0.9	0.98	0.30	0.95	0.33	0.97	0.32	58	1.00

**Table 4 ijerph-17-05707-t004:** Overview of the selected sub-populations with first-level events.

			#Reports in EHR			
Population	#Events	#Cases	Radiology	Pathology	MDO	Surgery
SVOB	17,677	5793	10,429	6987	199	62
NoSVOB	26,542	6427	15,254	10,208	784	296
Age ≥50	31,157	7740	18,132	12,894	819	312
Age <50	12,062	4480	7551	4,301	164	46
Birad12	23,393	8612	15,356	7874	131	32
Birad3-6	20,019	3365	9805	9041	849	324

**Table 5 ijerph-17-05707-t005:** Overview of the selected sub-populations with second-level events.

			#Sub-Levels in EHR							
Population	#Events	#Cases	Diagn	Biopsy	Other	Cyto	Histo	Benign	Maligne	MDO
2-Birad12	23,393	8612	15,141	204	11	1334	6540	1	31	131
2-Birad3-6	20,019	3365	9386	410	9	1205	7836	8	316	849

**Table 6 ijerph-17-05707-t006:** Population comparison. Reporting Jaccard similarity of sets of patient ids in the two groups. Note that the population is independent of the number of events considerd, i.e., cases Birad12 and 2-Birad12 are equivalent in terms of patients.

Group 1	Group 2	Jaccard Similarity
SVOB	NoSVOB	0.00
Age ≥50	Age < 50	0.00
Birad12	Birad3-6	0.00
NoSVOB	Age < 50	0.30
SVOB	Birad12	0.42

**Table 7 ijerph-17-05707-t007:** Process comparison on ZGT. Visual impression (Visual), conformance checking metrics (F—Fitness, P—Precision) when group 1 event log is checked against the process model (PM) constructed for group 2 and vice versa. Graph similarities (GED—graph edit distance, FS—feature-based similarity)

Populations			Conformance Checking						Graph Sim.	
			Log1-PM2		Log2-PM1		Average			
Group 1	Group 2	Visual	$F_{12}$	$P_{12}$	$F_{21}$	$P_{21}$	$\bar{F}$	$\bar{P}$	GED	FS
SVOB	NoSVOB	1.6±0.8	1.00	0.63	0.71	0.84	0.86	0.74	39	0.97
Age ≥50	Age < 50	2.2±1.1	0.69	0.86	1.00	0.52	0.85	0.69	34	0.96
Birad12	Birad3-6	1.9±0.9	0.82	0.59	0.57	0.73	0.70	0.66	36	0.90
NoSVOB	Age < 50	2.6±0.9	0.70	0.86	1.00	0.60	0.85	0.73	39	0.95
SVOB	Birad12	0.0±0.0	0.78	0.75	0.81	0.75	0.80	0.75	4	0.98
2Birad12	2Birad3-6	3.5±0.9	0.80	0.56	0.56	0.52	0.68	0.54	69	0.96

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
