# Peer review of "Automatic Process Comparison for Subpopulations: Application in Cancer Care"

_ijerph, 2020, doi:10.3390/ijerph17165707_

Round 1

Reviewer 1 Report

The paper is very interesting and tackles the very relevant problem of comparing process models. In addition, the paper uses two real datasets to show the results and ground the rest.

The paper is well written and the results presented are interesting. The paper, however, suffers from a couple of methodological errors:

- The first important issue is that one of the measures is completely unreliable. Specifically, authors stated that the visual comparison was done with 3 co-authors of the paper. Having 3 data points is absolutely not enough. In addition, being the subjects also authors of the paper, their objectivity is (involuntarily) compromised. I strongly suggest to collect many more opinions (~20?) from persons who are not biased with the work.

- The second methodological issue I have is with the correlation analysis. First: why the Pearson's correlation? Is the normality of variables assumption met? I doubt that. Also, the correlation value is not enough: what is the significance of those correlations? Without this information, the risk of just random observations is too high.

- The fundamental problem with GED is the lack of semantics: whenever talking about process models it is hard to ignore the fact that these nodes have meanings. For example, mixing a XOR and and AND split can result in MANY activities MORE/LESS to be done, though the measure will indicate just 1. I think it should be important to discuss this matter a bit more in details. In the literature, people talks about "trace equivalence" of process models or bisimilarity. I understand that the idea is to compare the graphs, but I think that Petri nets or BPMN have to much semantics. What about comparing the reachability graphs instead?

- Finally, there is no discussion about limitations of the method and the paper but I think these should be clearly stated and discussed.

Author Response

Dear Reviewer,

Thank you for your letter and the opportunity to revise our paper on ‘Automatic Process Comparison for Subpopulations: Application in Cancer Care’. I have included your comments in the attachment and addressed each comment separately. The revisions have been approved by all authors. We hope the revised manuscript will better suit the International Journal of Environmental Research and Public Health but are happy to consider further revisions, and we thank you for your continued interest in our research.

Francesca Marazza

Reviewer 2 Report

The presented article is an extension of a paper from the PODS4H Workshop. It provides a good example of how to automatically compare generated processes from EHR data. It compares several process models using two automatic and one manual technique. The main contribution of the paper is testing multiple ways to automate this task and apply it in a complex scenario, as it is healthcare.

The paper is well written and provides a good background of the related work.
A clear method is stablished and two case studies are executed. One using the freely available MIMIC dataset and another using the ZGT Hospital Data. For each of these datasets, the data was extracted, the models were generated and the analysis was done. The results are very interesting and are provided with good details. Overall, the exploration of comparing healthcare models is an interesting topic because of the complexity of these models and the high variability present in them.

I wanted to read more in the discussion or conclusions about the complexity of healthcare processes discovered using process mining, and how it is necessary to just use a small sample to get readable and understandable models. When you start to use a high amount of cases, the process models starts to be an spaghetti process and this can be another issue that should be addressed when trying to compare models.

Two minor changes or comments:

- Why was the Jaccard similarity not used with the MIMIC data?
- Include what is the specific extension of this paper from the original one.

Author Response

(The authors gave the same response as above.)

Round 2

Reviewer 1 Report

The authors adequately fixed my previous comments and the paper resulted in a methodologically much stronger contribution now.